# Modelling livestock infectious disease control policy under differing social perspectives on vaccination behaviour

Edward M. Hill[1,2]*, Naomi S. Prosser[3], Eamonn Ferguson[4], Jasmeet Kaler[3], Martin J. Green[3], Matt J. Keeling[1,2], Michael J. Tildesley[1,2]

**1** The Zeeman Institute for Systems Biology and Infectious Disease Epidemiology Research, School of Life Sciences and Mathematics Institute, University of Warwick, Coventry, United Kingdom, **2** Joint UNIversities Pandemic and Epidemiological Research, https://maths.org/juniper/, **3** School of Veterinary Medicine and Science, Sutton Bonington Campus, University of Nottingham, Leicestershire, United Kingdom, **4** School of Psychology, University Park, University of Nottingham, Nottingham, United Kingdom

* Edward.Hill@warwick.ac.uk

**Data Availability Statement:** We were supplied premises-level records from the Cattle Tracing System and sheep keeper inventory by the Rapid Analysis and Detection of Animal-related Risks

## Abstract

The spread of infection amongst livestock depends not only on the traits of the pathogen and the livestock themselves, but also on the veterinary health behaviours of farmers and how this impacts their implementation of disease control measures. Controls that are costly may make it beneficial for individuals to rely on the protection offered by others, though that may be sub-optimal for the population. Failing to account for socio-behavioural properties may produce a substantial layer of bias in infectious disease models. We investigated the role of heterogeneity in vaccine response across a population of farmers on epidemic outbreaks amongst livestock, caused by pathogens with differential speed of spread over spatial landscapes of farms for two counties in England (Cumbria and Devon). Under different compositions of three vaccine behaviour groups (precautionary, reactionary, non-vaccination), we evaluated from population- and individual-level perspectives the optimum threshold distance to premises with notified infection that would trigger responsive vaccination by the reactionary vaccination group. We demonstrate a divergence between population and individual perspectives in the optimal scale of reactive voluntary vaccination response. In general, minimising the population-level perspective cost requires a broader reactive uptake of the intervention, whilst optimising the outcome for the average individual increased the likelihood of larger scale disease outbreaks. When the relative cost of vaccination was low and the majority of premises had undergone precautionary vaccination, then adopting a perspective that optimised the outcome for an individual gave a broader spatial extent of reactive response compared to a perspective wanting to optimise outcomes for everyone in the population. Under our assumed epidemiological context, the findings identify livestock disease intervention receptiveness and cost combinations where one would expect strong disagreement between the intervention stringency that is best from the perspective of a stakeholder responsible for supporting the livestock industry compared to a sole livestock owner. Were such discord anticipated and achieving a consensus view across perspectives desired, the findings may also inform those managing veterinary health policy the requisite

(RADAR) team at the Animal and Plant Health Agency (APHA). These data contain confidential information, with public data deposition non-permissible for socioeconomic reasons. For data access, the RADAR team at APHA can be contacted on RADAR@apha.gov.uk. All other data utilised in this study are publicly available, with relevant references and data repositories stated within the main manuscript. The code repository for the study is available at: https://github.com/EdMHill/livestock_disease_control_differing_social_perspectives. Archived code at time of publication: https://doi.org/10.5281/zenodo.5730226.

**Funding:** EMH, NSP, EF, JK, MJG, MJK, MJT were supported by the Biotechnology and Biological Sciences Research Council [grant numbers: BB/S01750X/1, BB/5016341/1]. The funders had no role in study design, data collection and analysis, decision to publish, or preparation of the manuscript.

**Competing interests:** The authors have declared that no competing interests exist.

reduction in intervention cost and/or the required extent of nurturing beneficial community attitudes towards interventions.

## Author summary

The COVID-19 pandemic has shown how crucial human behaviour is in controlling the spread of an infectious disease. The same is true of livestock, where farmer behaviour is critical to reduce the spread of an infection to enhance animal welfare and reduce economic losses. An ongoing concern for livestock owners is therefore ensuring they have adequate disease management procedures. However, what an individual farmer considers an appropriate way to control an infection in their own livestock may not be the best way to prevent an infection for every farmer's livestock in the population. We describe a mathematical model combining epidemiological and behavioural elements to study the tension between individual and population-level control of livestock diseases. Applied to representative livestock systems in two counties in England (Cumbria and Devon), and splitting farmers into three types of vaccine behaviour groups (precautionary, reactionary, non-vaccination), we show what individual farmers see as an effective way to reduce infection is not the same as would benefit every farmer. The preferred response to protect every farmer's livestock is to encourage wider uptake of reactive vaccination, whereas optimising the spatial extent of reactive vaccination for the average individual increases the chance of larger disease outbreaks.

## Introduction

Infectious diseases in livestock are a blight on the agricultural industry, compromising animal welfare and reducing the profitability of affected animals. For example, estimates of the economic impact of bovine viral diarrhoea (BVD) ranges from £0 to £552 per cow per year [1], with an annual cost in Great Britain estimated in 2005 to be as high as £61M per year [2]. Above direct disease costs, imposition of controls aiming to curb the spread of infection can result in additional economic impacts, such as the cost of implementing the intervention, loss of income due to trade bans and fulfilling compensation arrangements [3]. Around £100 million is being spent annually in the UK on bovine tuberculosis (bTB) controls, encompassing disease surveillance, monitoring and non-genetic control (controls that do not include genetic contact tracing; genetic contact tracing aims to identify the source of bovine tuberculosis outbreaks by examining the DNA of the disease-causing bacteria sampled from cows and badgers that had tested positive) [4]. Another significant economic disease for the UK livestock industry is foot-and-mouth disease (FMD), due to its ability to spread very rapidly and its profound effect on productivity [5], though subsequent analysis of the 2001 UK FMD epidemic has highlighted how movement and export bans had the greatest economic consequences [6].

Disease management decisions by livestock owners can be influenced by several psychosocial components (attitudes and behaviours) [7]. Attempting to balance multiple factors (epidemiological, economic, animal welfare) can cause consequential tensions in the approach to disease control when inaction may be the most cost-effective strategy for an individual farmer, even though this could lead to local protracted epidemics. For example, dependent upon the disease that is circulating, an individual farmer may accept the risk of infection rather than proactively cull their livestock; yet culling may be a powerful tool to prevent large-scale

outbreaks of some diseases. Elicitation of cattle farmer psychosocial beliefs have identified multiple factors as having an association with proactive undertaking of control measures, which include the importance of veterinarians as a source of information, a lack of trust in other farmers, the individual experiences of farmers, and the knowledge and understanding of how and why to control disease and availability of time and money to enact the control [8, 9]. To formulate livestock disease intervention policies that will be highly adhered to by all stakeholders, there is a need to quantify, predict and hence mitigate these inevitable tensions between local (farmer-led) and global (regionally- and nationally-enforced) perspectives of control.

Mathematical models provide a viable means of extrapolating from known epidemiological behaviour, incorporating inevitable parameter uncertainty and integrating the non-linear dynamics, to produce predictions of how controls will impact the level of disease. The first large-scale use of epidemiological modelling to influence livestock policy decisions arose during the 2001 FMD outbreak, shaped by a need to bring the epidemic under rapid control [6, 10, 11]. By taking such models that encapsulate epidemiological processes and subsequently iterating upon them to mechanistically combine disease spread and farmer behaviour in a cohesive framework, we can produce a tool capable of assessing the differences between population-level and individual-level optimisation of controls.

There has been a call by the infectious disease modelling community, in the human public health field, for greater incorporation of behavioural changes among infectious disease dynamic models [12]. Likewise, there has been a growing literature of livestock disease modelling in recent years that explicitly includes the impact of behaviour on infection and control policy [13]. The majority of these prior works have used agent based models [14–16], with it previously demonstrated that the inclusion of individual heterogeneity in biosecurity positions within swine and cattle production systems can significantly affect epidemic dynamics [14, 16]. Human-behaviour factors need to be considered for improved epidemic forecasting. One of the intents of this study is to use a generic model setup, with the premises (farm) being the epidemiological unit of interest. Parsimonious models have the considerable advantage that any insights gained are unlikely to depend on the precise assumptions; they can provide a stepping stone to understanding more disease specific models and being less cumbersome means they can be a vital contributor in health emergency situations, where policy processes are fast moving and presentation of findings before a policy decision is taken is key [17].

In this paper, we explore the optimal intervention response to a disease outbreak amongst livestock, caused by pathogens with differential speed of spread, when viewed through two differing social perspectives: a population-level, taking the viewpoint of an entity who wants to minimise costs for the overall population when considered as a collective; an individual-level, minimising the cost across the population when all individuals judged costs from a personal point of view (specifically the additional costs suffered as a result of the use/non-use of an available disease control measure being a suboptimal decision). We aim to develop an intuitive understanding of when we may expect a deviation in the disease management practices discerned as being best between the two social perspectives. The behavioural trait we focus on is responsive uptake of vaccination based on the perceived infection risk (where we use the distance away from a premises notifying the presence of infection as a proxy measure). To capture known heterogeneities in vaccination attitudes within a population, with distinct sub-groups emerging who take very different and distinct vaccination decisions [18, 19], we incorporated these behavioural elements into a mathematical model of infectious disease transmission. Informed by spatial and livestock demographic data for two counties in England (Cumbria and Devon), we exemplify on realistic livestock systems discords in livestock disease management views being common. The complex interaction between relative cost of vaccination and

vaccine use stance means that, taking either perspective, there are circumstances where controls would be activated over a wider spatial extent than what may be favourable when taking the other perspective.

## Methods

We simulated an epidemic process over spatial landscapes of farms (specifically premises with cattle and/or sheep in two counties of England) and ascertained the impact on the optimal control policy that minimised cumulative costs (from infection and application of interventions) when adopting a population or individual perspective. Through the Methods we detail: (i) the data sources used to parameterise the spatial locations and livestock demography, (ii) the epidemiological model framework that was conceptually based on FMD-like transmission with no long-range movement of animals, (iii) parameterisation of the epidemiological model, (iv) how we implemented vaccination in the model, (v) our assessment of costs from differing perspectives (population versus individual), and (vi) the simulation protocol used to assess the scenarios of interest.

### Data description

**Spatial locations.** To explore the sensitivity of the identified optimal intervention strategies to spatial and demographic characteristics, we performed our analysis for two different counties in England.

We selected Cumbria and Devon, with these counties being spatially distant (Fig 1A), both having a high density of cattle farms (being badly affected by FMD in 2001) and having a similar area (Cumbria: 6768km$^2$ [20], Devon: 6707km$^2$ [21]).

**Livestock data.** With our simulated pathogen having transmission characteristics akin to FMD (details in the subsections "Epidemiological model" and "Model parameterisation"), following previously used livestock disease model frameworks for FMD in the UK we included two livestock types, cattle and sheep [11, 22–26]. We sourced premises-level estimates for cattle herd and sheep flock sizes in 2020 within Cumbria and Devon from two datasets.

For cattle, we used the Cattle Tracing System database to procure average herd sizes throughout 2020 for each premises. The Cattle Tracing System contains virtually complete records of the births, deaths, and movements of individual cattle in Great Britain since 2001 [27]. For sheep flock sizes, we used estimates from December 2020 that were submitted to the annual sheep keeper inventory [28].

**Description of our study dataset.** Our county data contained premises that recorded keeping either (or both) cattle or sheep. With these criteria we retained 3784 premises from Cumbria and 5343 from Devon (Fig 1B). The density of premises was lower in Cumbria (0.56 premises per km$^2$) than Devon (0.80 premises per km$^2$).

From the total number of premises that had either cattle or sheep present, Cumbria had a lower percentage of premises compared to Devon with just cattle (28.1% vs 41.9%), and a similar percentage of premises with sheep but no cattle (26.6% vs 29.2%). On the other hand, Cumbria had a higher percentage of premises that had both cattle and sheep present (1717 out of 3784 premises, 45.4%) compared to Devon (1542 out of 5343 premises, 28.9%) (Fig 1C).

Checking for similarities in the premises-level livestock population distributions between Cumbria and Devon, we found a dependence on the livestock type (Fig 1D). Note that we calculated these summary statistics using only those premises where the given livestock type (cattle or sheep) was present. The distribution of cattle herd sizes were comparable (Cumbria: median 84, interquartile range (IQR) 25–202, 97.5th percentile 681; Devon:

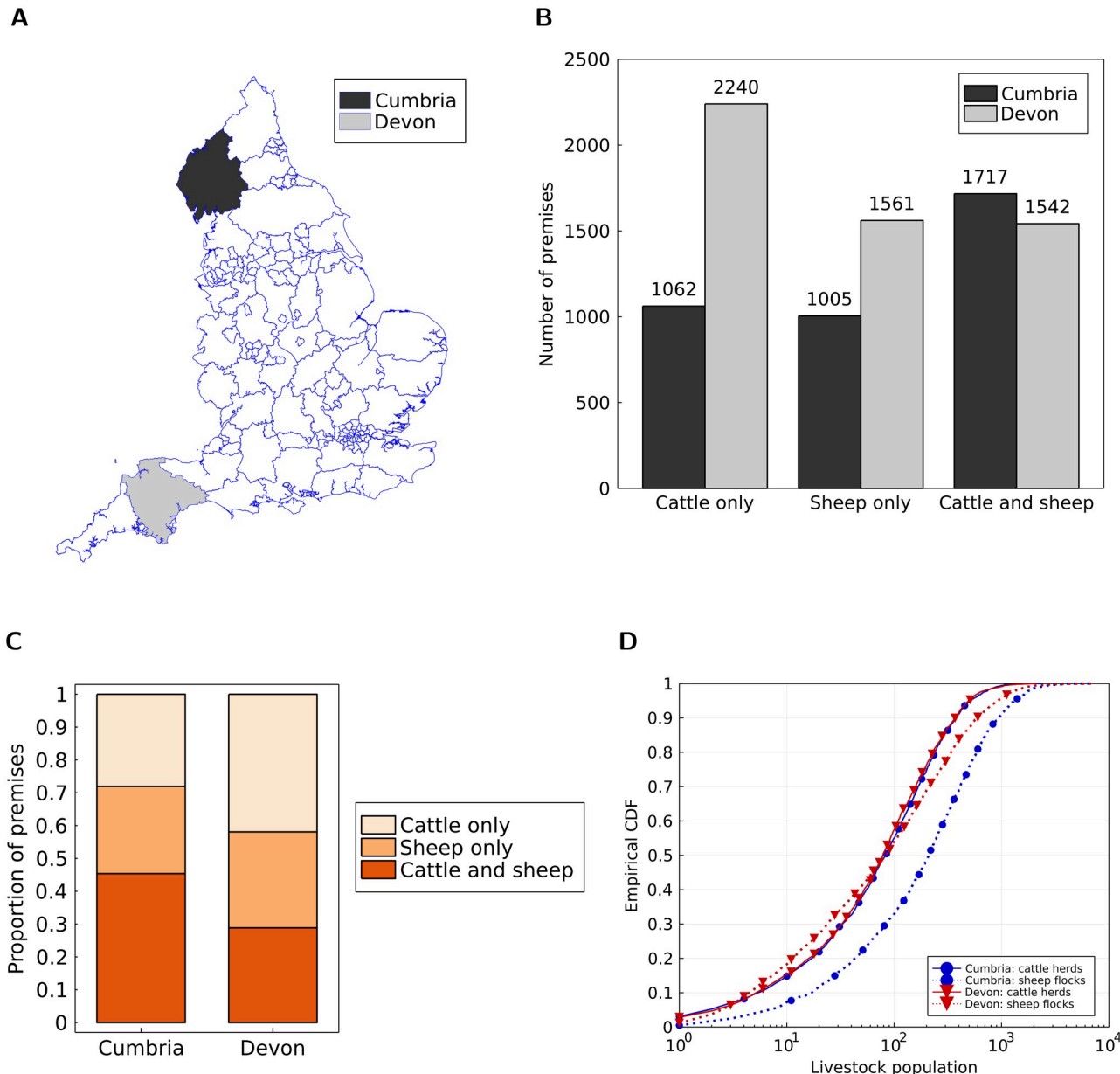

**Fig 1. Livestock demographics of premises in our Cumbria and Devon county datasets.** Our county datasets contained premises that had at least one positive count of cattle (estimates from the 2020 CTS data) or sheep (estimates from the December 2020 sheep inventory). **A** Locator map for Cumbria (shaded black) and Devon (shaded grey). The base layer of the map was made available by the Office for National Statistics [29]; URL: https://www.arcgis.com/home/item.html?id=aff50e8d15364a7b82c62c14861eb240. Source: Office for National Statistics licensed under the Open Government Licence v.3.0. Contains OS data Crown copyright and database right 2022. We display per county: **B** the number of premises with cattle only, sheep only or both cattle and sheep; **C** the proportion of premises with cattle only, sheep only or both cattle and sheep. **D** Empirical cumulative distribution functions of cattle herd size (solid lines) and sheep flock size (dashed lines) for Cumbria (blue lines with circle markers) and Devon (red lines with inverted triangle markers). In panel **D**, we limited the calculation of the empirical cumulative distribution functions to only include premises where the given livestock type (cattle or sheep) was present.

median 79, interquartile range (IQR) 24–191, 97.5th percentile 633). In contrast, sheep flocks were typically greater in Cumbria (median 210, interquartile range (IQR) 61–493, 97.5th percentile 1715) than Devon (median 84, interquartile range (IQR) 18–270, 97.5th percentile 1223).

## Epidemiological model

To reflect the inherent spatial nature of transmission of livestock infection, our model of the infectious disease dynamics consisted of a spatial model for a spatially spreading pathogen with a latent stage and presymptomatic infectious stage (an infection history with a likeness similar to that of FMD). We express that although the scenario of a pathogen spreading fairly rapidly between farms is relatively rare in UK livestock farming, it is an important scenario to consider as a pathogen spreading rapidly between farms has the potential to cause colossal negative impacts.

Our formulation was a stochastic premises level Susceptible-Exposed-Infectious-Removed (SEIR) model with discrete daily time steps and a daily force of infection, $\lambda_{ij}$, between infectious farm $i$ and susceptible farm $j$. With our epidemiological unit being at the level of the premises, we inherently assumed that once infection had entered the livestock population on a premises the within-premises disease spread occurred rapidly, leaving all livestock infected.

The force of infection between two premises depended on the dispersal kernel $K$, which was a function of the Euclidean distance between premises $i$ and $j$ ($d_{ij}$), as well as the transmissibility of farm $i$ ($T_i$) and the susceptibility of farm $j$ ($S_j$). The daily force of infection was given by:

$$\lambda_{ij} = T_i S_j K(d_{ij})$$

It follows that the daily probability of a susceptible farm $j$ becoming infected by farm $i$ is obtained by:

$$p_{ij} = 1 - \exp(\lambda_{ij})$$

Following Tildesley *et al.* [24], we modelled transmissibility and susceptibility of farms as a nonlinear function of the number of cattle and sheep present on the premises as

$$T_i = \xi_c N_{c,i}^{\psi_c} + \xi_s N_{s,i}^{\psi_s},$$
$$S_i = \zeta_c N_{c,i}^{\phi_c} + \zeta_s N_{s,i}^{\phi_s},$$

with $N_c$ and $N_s$ the number of cattle and sheep at the premises, and the following livestock-specific parameters: per animal transmissibility ($\xi_c, \xi_s$), per animal susceptibility ($\zeta_c, \zeta_s$), infectious population exponent ($\psi_c, \psi_s$) and susceptible population exponent ($\phi_c, \phi_s$).

## Model parameterisation

Once a premises was infected, we assumed a latent period of five days (based on epidemiological and veterinary records from the 2001 UK FMD outbreak [11]). Afterwards, the entire livestock population at that premises was considered infectious for a period of eight days (days 6–13 after infection). We assumed all infected premises provided notification of infection nine days after the initial infection event, meaning there was no under-reporting of infection, but there was a four day delay between the premises becoming infectious and subsequent notification of infection. At the end of the infectious period (13 days after infection) the premises and its livestock were considered removed from the population.

For simplicity, we used sheep and cattle stratified epidemiological parameter estimates inferred from the 2001 UK FMD epidemic for Cumbria [24], which informed: per animal transmissibility ($\xi_c, \xi_s$), per animal susceptibility ($\zeta_c, \zeta_s$), infectious population exponent ($\psi_c, \psi_s$) and susceptible population exponent ($\phi_c, \phi_s$).

To scale the risk of transmission with distance, through the transmission kernel $K$, we applied a power-law transmission kernel with best fit values as inferred in [30]. In detail, the

transmission kernel has the functional form:

$$K(d) = \frac{k_1}{1 + \left(\frac{d}{k_2}\right)^{k_3}},$$

with $d$ the distance between the two premises, $k_2 = 1600$ is the scale parameter and $k_3 = 4.6$ is the shape parameter. The parameter $k_1 = 0.08912676$ is the normalising constant that scales the function so that

$$\int_0^\infty 2\pi r K(r) dr = 1,$$

where $r$ is equivalent to $d$, the distance between premises, in the spatial kernel equation. We recognise that this parameterisation of the transmission kernel is based on cattle disease outbreaks in the USA, whereas our spatial landscape and livestock epidemiological parameters are informed by data from the UK. We initially investigated using a parameterisation of the transmission kernel inferred from 2001 UK outbreak data. However, with our livestock population estimates being from 2020 and the changes in demography that have occurred during those two decades (fewer premises and generally larger herd/flock sizes), using a parameterisation of the transmission kernel based on 2001 UK outbreak data would not necessarily result in epidemic outbreaks. On the other hand, for demonstrating the utility of the model framework, using the previously described parameterisation for the transmission kernel (with parameters inferred by Buhnerkempe *et al* [30]) was ideal for the investigative purposes of our study as it produced a range of epidemic outcomes in both county settings.

We provide an overview of the epidemiological model values in Table 1. In addition to our default model parameterisation, as a sensitivity analysis we considered an alternative parameterisation of the pathogen characteristics. Explicitly, the 'alternate' pathogen had the same reproduction number as the pathogen in our main analysis, but it had an elongated infectious period (from eight days to 32 days), in combination with a factor of four reduction in the per animal transmissibility parameters (see the values in parentheses in Table 1).

**Table 1. Summary of parameter notation and descriptions.** All values are for the default pathogen characteristics, except those stated within parentheses that correspond to the alternate pathogen.

| Symbol | Description | Value |
|---|---|---|
| $\lambda_j(t)$ | Infectious pressure on susceptible farm $j$ at time $t$. | Variable |
| $t_{\text{incub}}$ | Time elapsed until end of incubation period (relative to the day of infection) | 5 days |
| $t_{\text{notif}}$ | Time elapsed until notification (relative to the day of infection) | 9 days |
| $t_{\text{removal}}$ | Time elapsed until removal (relative to the day of infection) | 13 days (37 days) |
| $N_{c,j}, N_{s,j}$ | Number of cattle and sheep (respectively) on premises $j$. | Variable |
| $\xi_c, \xi_s$ | Transmissibility of cattle ($c$) and sheep ($s$). | 8.2e-4, 8.3e-4 (2.05e-4, 2.075e-4) |
| $\zeta_c, \zeta_s$ | Susceptibility of cattle ($c$) and sheep ($s$). | 5.7, 1 |
| $\psi_c, \psi_s$ | Exponent on the cattle ($c$) and sheep ($s$) population on an infectious farm, for calculating the infectious pressure. | 0.42 0.49 |
| $\phi_c, \phi_s$ | Exponent on the cattle ($c$) and sheep ($s$) population on a susceptible farm, for calculating the infectious pressure. | 0.41, 0.2 |
| $k_1$ | Transmission kernel normalisation constant. | 0.08912676 |
| $k_2$ | Transmission kernel distance length scale scaling ($k_2$). | 1600 |
| $k_3$ | Transmission kernel exponential parameter on the distance component. | 4.6 |

## Implementation of vaccination

To reduce system complexity we considered a single type of intervention, namely vaccination. Expansions in the intervention space can be conceived, such as considering other methods (culling or enhanced biosecurity, for example) and a package of multiple interventions, though are beyond the scope of this study.

Furthermore, within vaccination parameter space we could conceivably include a dimension corresponding to a range of postulated vaccine efficacies. We present here an idealised situation where the vaccine was 100% effective in blocking infection, though the protective qualities of vaccination were not assumed to be instantaneously gained post receiving the vaccine. We included a lag for the vaccine inducing an immune response of 4–6 days, based on measures of FMD virus titres in milk from inoculated cows in the days post-inoculation [31]).

These vaccine profile choices meant we could focus on discerning if there were disparities in the optimal disease management approach between population and individual perspectives, given the availability of a very highly effective intervention. Incorporating imperfect intervention effectiveness is another worthy direction of further study.

To enable assessment of the role of behaviour on epidemic outcomes, we stratified the population of farmers into three vaccination groups: 'precautionary', 'reactionary' and 'non-vaccinators'. Farmers in the 'precautionary' group had all their livestock (both cattle and sheep) set as being vaccinated and fully protected from the beginning of the simulation. Farmers in the 'reactionary' group would vaccinate all livestock on their premises once a given 'risk' measure was satisfied. For this study the risk measure that triggered vaccination was there being a notified premises (confirming presence of infection, with there being a four day delay between the premises becoming infectious and infection being reported) within a distance $d$ km of their premises. Finally, farmers in the 'non-vaccinators' group did not vaccinate their livestock at any time, irrespective of the outbreak situation. We limited our model simulations to have no more than our three vaccination groups of 'precautionary', 'reactionary' and 'non-vaccinators' on the grounds of retaining model parsimony, with these three sub-groups reflecting the types of heterogeneity observed in vaccination behaviour and attitudes [19, 32].

Notified premises did not undergo vaccination. Vaccination was ineffective if the vaccination was administered to a latent infected or infectious (but not yet notified) animal.

## Costs under differing perspectives

For our analyses we applied a per animal cost of infection ($C_I$) and cost of vaccination ($C_V$), assuming the same per animal cost for cattle and sheep. We recognise this simplifying assumption may not be commonly borne out for pathogens affecting multiple livestock species, with the severity of disease and the costs to apply a given intervention being dependent on the type of livestock. The framework presented here has the flexibility to be tailored to an epidemiological context with these cost heterogeneities, with our focus being on attaining general principles from a parsimonious model approach.

Given an objective function, we can calculate an economically optimal threshold for different vaccination costs relative to infection costs (setting $C_I = 1$). However, the objective is typically subjective and there may be differences in opinion as to what the objective should ultimately be [33]. Here, we compare the optimal reactive vaccination ring size from population- and individual-level perspectives.

**Population perspective.**  The population perspective took the viewpoint of a stakeholder responsible for supporting and protecting the livestock industry as a whole, who would therefore seek to minimise costs across all livestock owners. Consequently, the optimal notification distance threshold $d$ from the perspective of the overall population simply minimised the

combined cost of vaccination and disease:

$$\min_d((C_V \times N_V) + (C_I \times N_I)) \tag{1}$$

where $N_V$ corresponds to the total number of livestock vaccinated and $N_I$ corresponds to the total number of livestock infected.

**Individual perspective.**   From the individual perspective, the optimal notification distance threshold $d$ minimised the cost across the population when all individuals judged costs from a personal point of view. In practice, this meant accounting for the four possible combinatorial outcomes of vaccination and infection status, with the costs associated with each outcome coinciding with the additional spend incurred by the farmer due to taking the 'incorrect' intervention option (whether that be vaccinating their livestock or leaving their livestock unvaccinated).

We let $!V\&!I$ denote entities that were not vaccinated and not infected, $!V\&I$ entities that were not vaccinated and infected, $V\&!I$ entities that were vaccinated and not infected, and $V\&I$ entities that were both vaccinated and infected. We attributed the following costs to the four possible combinatorial outcomes of vaccination and infection status:

1. $\text{Cost}_{!V\&!I}$: farmers who did not vaccinate and whose livestock did not become infected accrued no costs (per animal cost: 0);

2. $\text{Cost}_{V\&!I}$: for farmers who vaccinated their livestock and the livestock did not become infected, the cost reflected the probability of the livestock not being infected in any case if they had been left unvaccinated (per animal cost: a proportion of $C_V$, with the scaling linked to the probability of the livestock not being infected under the counterfactual of not having been vaccinated);

3. $\text{Cost}_{!V\&I}$: for farmers who did not vaccinate their livestock and whose livestock became infected, the costs represented the extra loss suffered above the cost of vaccinating (per animal cost: $C_I - C_V$);

4. $\text{Cost}_{V\&I}$: for farmers that vaccinated their livestock but whose livestock still became infected, the cost represented the extra infection-caused loss above the cost of vaccinating (per animal cost: $C_I$).

Note that despite the fully efficacious vaccine assumption, the vaccine could be administered to an infected population during its latent phase (thus prior to onset of symptoms and subsequent notification of infection). Therefore, in these circumstances it was feasible for the livestock at a premises to be vaccinated but still become infected.

Following completion of the outbreak, we defined $N_{!V\&!I}$ as the number of livestock that were not vaccinated and were not infected, $N_{V\&!I}$ as the number of livestock that were vaccinated and not infected, $N_{!V\&I}$ as the number of livestock that were not vaccinated and that were infected, and $N_{V\&I}$ as the number of livestock that were vaccinated but still became infected. From the individual perspective the objective function to be optimised, in full, was:

$$\min_d(\text{Cost}_{!V\&!I} + \text{Cost}_{V\&!I} + \text{Cost}_{!V\&I} + \text{Cost}_{V\&I}),$$

$$\text{where } \text{Cost}_{!V\&!I} = 0,$$

$$\text{Cost}_{V\&!I} = C_V \times \left( \sum_{i \in \hat{V}} \exp(-\sum_{t > T_{V_i}} \lambda_i(t)) \right),$$

$$\text{Cost}_{!V\&I} = (C_I - C_V) \times N_{!V\&I},$$

$$\text{Cost}_{V\&I} = C_I N_{V\&I}.$$

with $\hat{V}$ representing premises whose livestock were vaccinated by the end of the outbreak, $T_{V_i}$ the time of vaccination for premises $i$ and $\lambda_i(t)$ the force of infection against entity $i$ at time $t$. The composite term $\exp(-\sum_{t>T_{V_i}} \lambda_i(t))$ in $\text{Cost}_{V\&!I}$ corresponds to the probability of livestock at premises $i$ not being infected from the time they underwent vaccination ($T_{V_i}$) onward, under the counterfactual scenario of the livestock remaining unvaccinated for the entire duration of the outbreak.

As instances of livestock at premises being unvaccinated and not becoming infected had no personal cost to the farmer ($\text{Cost}_{!V\&!I = 0}$), and we fixed $C_I = 1$ so our analysis took a perspective of investigating the relative cost of vaccination (versus cost of infection), from the individual perspective the objective simplified to:

$$\min_d(\text{Cost}_{V\&!I} + \text{Cost}_{!V\&I} + \text{Cost}_{V\&I}),$$

$$\text{where } \text{Cost}_{V\&!I} = C_V \times \left( \sum_{i \in \hat{V}} \exp(-\sum_{t>T_{V_i}} \lambda_i(t)) \right),$$

$$\text{Cost}_{!V\&I} = (1 - C_V) \times N_{!V\&I},$$

$$\text{Cost}_{V\&I} = N_{V\&I}.$$

(2)

## Simulation outline

To mimic a situation of an emergent outbreak beginning in a spatially localised area from a low case level, in each simulation replicate we seeded infection in a randomly selected cluster of five premises (we selected one premises at random and found the four premises that were closest in terms of euclidean distance). A replicate terminated upon there being no premises in an infected state. For each vaccination group scenario we optimised the size of the reactive vaccination ring radius ($d$km) for the population perspective (Eq (1)) and the individual perspective (Eq (2)), with respect to a relative cost of vaccination compared to infection, $C_V$.

We first analysed a scenario where all farmers were in the 'reactionary' vaccination group. We carried out 500 outbreak replicates for 11 different reactionary vaccination ring distances, with $d$ taking on values 0km (equivalent to no reactionary vaccination) to 10km at 1km increments; as a consequence of our choice of 1km increments it was not feasible for intermediate distances between integer values to be optimal. Iterating over a set of relative cost of vaccination values in turn, $C_V \in \{0, 0.01, \ldots, 0.99, 1.00\}$, we discerned for each $C_V$ the input values for $d$ that were optimal from the population perspective (optimising Eq (1)) and individual perspective (optimising Eq (2)), respectively.

To explore the sensitivity of different behaviour profiles on the control discerned as optimal, we next considered a collection of scenarios with the population split between the three vaccination groups: 'precautionary', 'reactionary' and 'non-vaccinators'. Stratifying group occupancy to a resolution of 5% (a subjective choice that ensured the total required computational time for running the collection of scenarios was manageable, whilst providing a resolution that would be capable of revealing trends between vaccine stance group composition and epidemiological outcomes), we considered unique combinatorial combinations of occupancy across the three groups that summed to unity, giving 231 scenarios. Once more, within each scenario we performed 500 outbreak replicates for the 11 reactionary vaccination ring distances $d$ ranging from 0km to 10km (with a 1km increment). For the relative cost of vaccination we considered six values, $C_V \in \{0.01, 0.20, 0.40, 0.60, 0.80, 1.00\}$.

To gather insight on the burden of infection and the likelihood of containing an outbreak, in our initial analysis we inspected the percentage of premises infected, while in the latter analysis studying sensitivity of behaviour profiles we computed the percentage of simulation replicates that resulted in 25 or more infected premises. In all scenarios we also tracked the expected percentage of premises undergoing vaccination.

For our simulation procedure we employed the Sellke construction [34]. A desirable characteristic of the Sellke construction is encoding the inherent randomness of an epidemic realisation at the beginning of the simulation with a random vector $Z$ of Exp(1) distributed resistances. We achieve a direct comparison of a collection of control measures via the Sellke method by matching values of $Z$ at the epidemic outset. We performed all model simulations and produced plots in Julia v1.6, with the exception of the ternary plots that were produced in Matlab R2021b using the Ternary Plots package from the MATLAB Central File Exchange [35]. The code repository for the study is available at: https://github.com/EdMHill/livestock_disease_control_differing_social_perspectives.

## Results

### All reactionary stance

Analysing the scenario where all farmers were in the reactionary vaccination group, across the tested relative cost of vaccination range ($C_V \in \{0, 0.01, \ldots, 0.99, 1.00\}$) we found adopting the population perspective led to wanting a marginally wider reactionary vaccination zone (Fig 2).

Delving into the outcomes for Cumbria, for a near zero cost of vaccination ($C_V \leq 0.02$) a wide distance threshold of 5km or more would be enacted given either perspective. For the majority of $C_V$ values, there was a 1km separation in the notification distance between the population and individual perspectives; when $C_V$ was in the range 0.2–0.9 the optimal reactive vaccination distance remained consistent, at 4km under the population perspective and 3km under the individual perspective. The gap in the optimal strategy broadened when the cost of vaccination per animal was similar to the cost of infection per animal. In particular, for $C_V \geq 0.95$, if a farmer were to take an individual perspective it would never be worth carrying out vaccination, whilst from the population perspective the optimal notified premises distance within which vaccination would be triggered remained at 4km (Fig 2A).

For Devon, there were similar relationships between the optimal reactive vaccination strategy, relative cost of vaccination and the cost perspective (Fig 2B). However, from either perspective and for a specified $C_V$, the notification of infection distance threshold $d$ that minimised the cost for Cumbria matched or was marginally greater than the strategies identified as optimal for Devon. For example, as $C_V$ approached 1 the risk threshold from the population perspective was 3km (compared to 4km for Cumbria). Furthermore, from the individual perspective the relative cost above which it was not worth carrying out any reactive vaccination was lower for Devon ($C_V \geq 0.87$).

The requirement for slightly broader vaccination coverage in Cumbria was a consequence of the typical outbreak being larger in comparison to Devon (Fig 2C and 2D). Nevertheless, irrespective of the spatial locale, the more stringent controls that optimised outcomes for the population typically curbed outbreaks, with the percentage of premises infected in Cumbria and Devon unlikely to exceed 2% or 1%, respectively. When $C_V$ was close to 1, the deviation in control strategy if pursuing the individual perspective (having no vaccination) caused striking disparities in the anticipated burden of infection. In the absence of any vaccination, for Cumbria the average outbreak infected nearly 20% of premises and there were instances of over 40% of premises being infected (Fig 2C), with Devon having a median of 5% premises infected and some replicates causing in excess of 30% of premises being infected (Fig 2D).

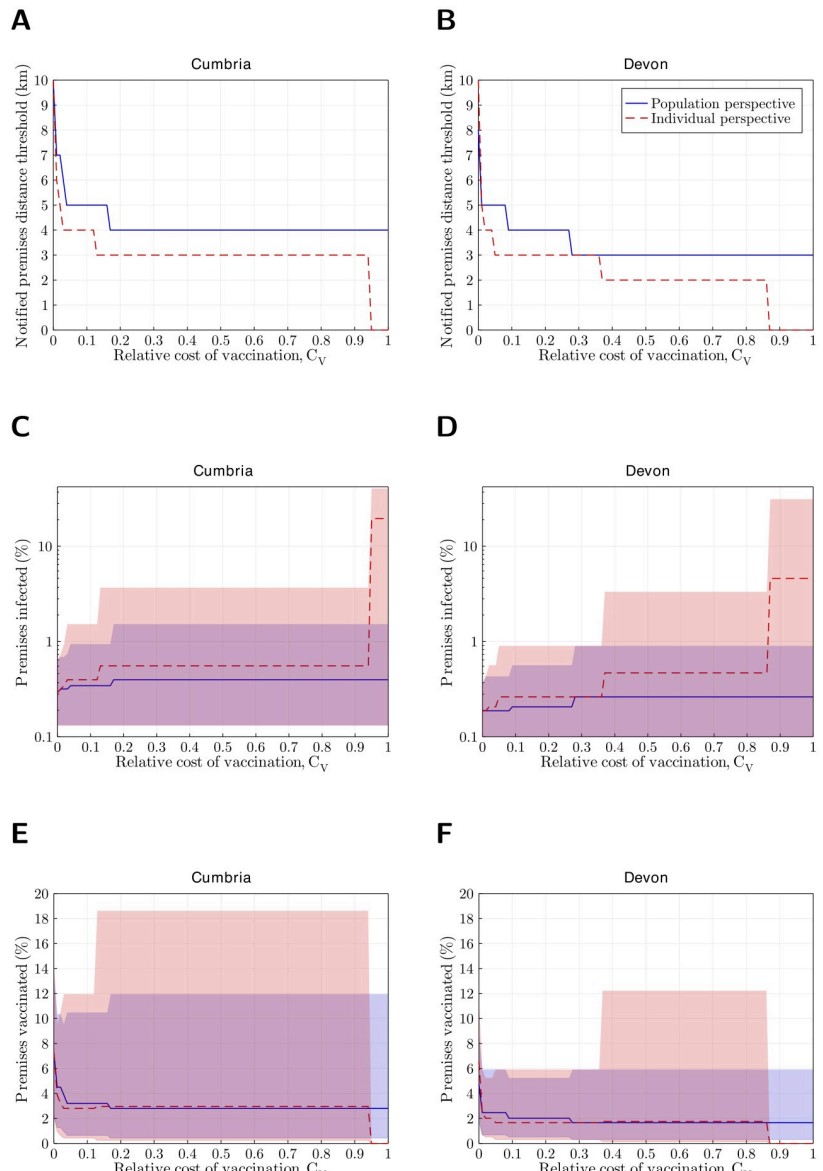

**Fig 2. For the 'default' pathogen, the reactive vaccination strategy that minimises the overall cost from either the population or individual perspectives, dependent upon the relative cost of vaccination versus infection.** We performed 500 replicates per scenario. Vaccination at a premises was triggered by notification of infection within the specified distance threshold (no premises were initialised as either having already been vaccinated or in the 'never vaccinate' group). We identified the lowest median cost (with there being a distribution of costs from the ensemble of stochastic simulations) across the control scenarios from a population perspective (blue solid line) and an individual (farmer) perspective (red dashed line) for: **A** Cumbria; **B** Devon. We recall that the possible reactive vaccination distance thresholds were 0km to 10km inclusive (in 1km increments). Therefore, as it was not possible for intermediate distances between these integer values to be optimal, we obtained step-like line profiles. Under the identified optimum reactive vaccination strategy, we show medians (lines) and 95% prediction intervals (shaded regions) for: **C,D** the percentage of premises infected (note the y-axis is on a log scale); **E,F** the percentage of premises vaccinated.

As a consequence of there being (on average) a higher amount of infection expected and wider zones to undergo reactive vaccination, the average amount of reactionary vaccination was higher in Cumbria than Devon. This relationship held irrespective of the relative cost of vaccination and the perspective taken (Fig 2E and 2F). With the exception of $C_V$ values close

to 0 or 1, the percentage of premises that underwent vaccination were similar under the optimal strategy from either perspective; approximately 3% for Cumbria (95% prediction intervals (PI): 0–12% for the population perspective, 0–19% for the individual perspective) and approximately 2% for Devon (95% PI: 0–6% for the population perspective, 0–12% for the individual perspective).

Were we instead facing the 'alternate' pathogen that had a lower transmission potential (per unit time) but a longer infectious period, there was closer agreement in the course of action to take from the population or individual perspective. Nonetheless, once more there were no instances of the individual perspective having a broader notification zone versus the population perspective (Fig A(a,b) in the S1 Text). Independent of $C_V$, the differing characteristics presented by such a pathogen meant that, in general, a reactive vaccination zone with a radius reduced by 1km compared to the 'default' pathogen was sufficient. Furthermore, the qualitative relationships between the percentage of premises infected and vaccination coverage against $C_V$ were similar for the 'alternate' pathogen as for the 'default' pathogen. Quantitatively, we observed reduced median values and narrower prediction intervals for both infection (Fig A(c,d) in the S1 Text) and vaccination (Fig A(e,f) in the S1 Text).

## Sensitivity to vaccine behaviour

We expanded the dimensionality of the system by amending the stance towards vaccination from a single type, 'reactionary', to three types, 'precautionary' (who were set to be vaccinated from the beginning of the simulation), 'reactionary' (who would vaccinate if the risk measure criteria was satisfied) and 'non-vaccinators' (irrespective of the epidemiological situation, would never vaccinate). We considered 231 vaccine stance group composition scenarios, evaluating the notification distance $d$km triggering vaccination for the reactionary group that minimised the cost from the population and individual perspectives with respect to a relative cost of vaccination $C_V \in \{0.01, 0.20, 0.40, 0.60, 0.80, 1.00\}$.

We initially scrutinise the outcomes for Cumbria. We remark that for the degree of correspondence (or lack thereof) in the optimal strategy from the population and individual perspectives, there was an interplay between $C_V$ and how the population was partitioned between the three behavioural types (Fig 3). There were parameter combinations, particularly when the relative cost of vaccination was small, where taking an individual perspective would result in the notification distance threshold (the 'reactionary' group risk measure to determine whether they should vaccinate their livestock) being higher compared to the population perspective. Specifically, in the regime where $C_V \leq 0.6$ and the 'precautionary' behaviour group comprised 50% or more of the population, there were behaviour type compositions where from the population perspective it would be preferable for the 'reactionary' group to not vaccinate (Fig 3B and 3C), whereas those who were 'reactionary' and taking an individual perspective would want to vaccinate were there a premises notifying infection nearby (Fig 3E and 3F). For $0.6 < C_V < 1$, the control response that would be endorsed from a population standpoint (versus the individual perspective) typically had a wider notification zone to trigger vaccination for those with a 'reactionary' vaccination stance (Fig 3G, 3H, 3J and 3K). When the cost of vaccination per animal was equivalent to the cost of infection per animal ($C_V = 1$), from the population perspective if the percentage of precautionary behaviour individuals was below 40% then there was a benefit to reactionary vaccination taking place (Fig 3I). In contrast, from the individual perspective vaccination offers zero cost savings when it costs as much to vaccinate each animal as the monetary loss suffered for each animal infected. Accordingly, the optimal strategy was no reactive vaccination (corresponding to 0km notification distance for all 231 vaccine stance group compositions, Fig 3L). In general, if everyone followed an individual perspective there

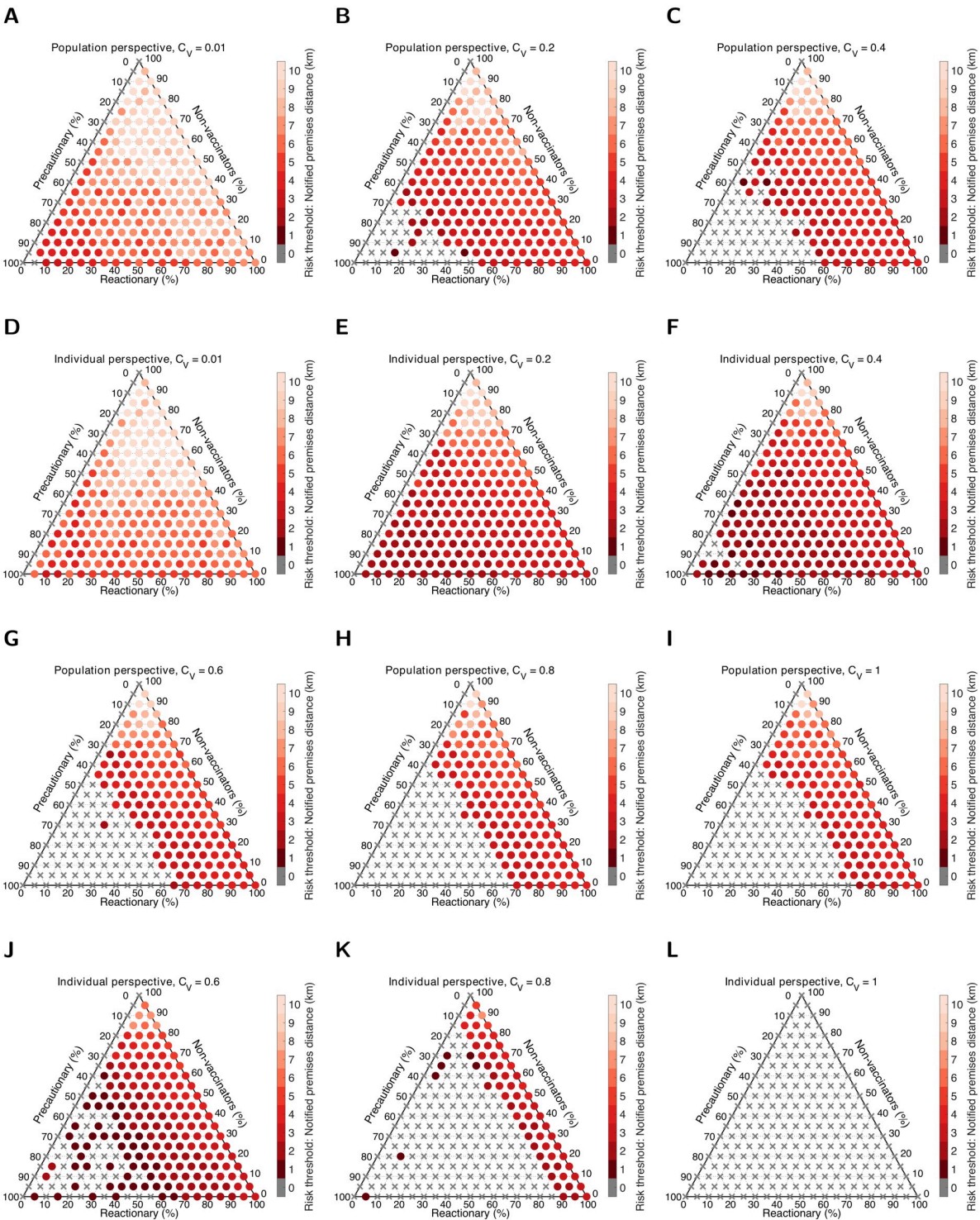

**Fig 3. Strategy that minimises overall cost in Cumbria, dependent on the perspective, relative cost of vaccination and vaccine group composition.** We performed 500 replicates per scenario. Shading corresponds to the notification distance risk threshold $d$ that returned the lowest median cost across the control scenarios (dark to light corresponding to increases in distance from 1km to 10km). Grey crosses correspond to no reactive vaccination occurring. The ternary plots display optimal strategies from the population and individual perspectives given a relative cost of vaccination, $C_V$, of: **A,D** 0.01; **B,E** 0.2; **C,F** 0.4; **G,J** 0.6; **H,K** 0.8; **I,L** 1. For an explanation on reading values on a ternary plot, see Fig B in the S1 Text.

was a reduced likelihood of curbing outbreak size, as measured by our measure of percentage of replicates resulting in 25 or more premises being infected. The risk of a larger outbreak was appreciably enhanced when the majority of the population were 'reactionary' and few were 'non-vaccinators' (lower right section of the ternary plots, Fig C in the S1 Text).

The general relationships between the relative cost of vaccination, vaccine group composition, cost perspective and the notified premises distance ascertained as optimal also held in the Devon setting (Fig 4). As for the all reactionary group scenario, there was usually a minor decrement in the notified premises distance compared to the equivalent scenario applied to Cumbria. As a result, we observe larger regions of vaccine behaviour type composition space where the optimal response of the 'reactionary' group was to not vaccinate (visualised by there being relatively more grey cross markers fanning out from the bottom left corner of the ternary plots). The trend of the vaccination effort in Devon not needing to be as intensive as Cumbria coincided with the region having less chance of outbreaks exceeding 25 or more premises infected; there was a greater likelihood of suppressing the initial cluster of infection in Devon than Cumbria (Fig C and D in the S1 Text). Median vaccination coverage was comparable in all scenarios for both regions (Fig E and F in the S1 Text).

For the 'alternate' pathogen, there were subtle shifts in the quantitative relationships between perspective taken, vaccination cost and behavioural characteristics across the population. There were fewer instances where those who were 'reactionary' and taking an individual perspective would want to vaccinate were there a premises notifying infection in the local neighbourhood, when from the population perspective it would be preferable for the 'reactionary' group to not vaccinate; we observed such outcomes when the cost of vaccination per animal was about half the cost of infection per animal ($C_V$ in the range 0.4–0.6). As $C_V$ increased the region of behaviour type space broadened where 'reactionary' vaccination would occur under the population perspective, but not when adopting an individual perspective (Fig G and H in the S1 Text). Qualitative relationships were similar for the 'alternate' pathogen, as for the 'default' pathogen, between $C_V$ and both our outbreak risk measure (percentage of replicates in which 25 or more premises were infected, Fig I and J in the S1 Text) and vaccination coverage (Fig K and L in the S1 Text).

## Discussion

What an individual may consider the best approach to control an emerging disease outbreak (i.e optimising costs from their individual perspective) may not align with what would be the most effective collective action (i.e optimising costs from a population perspective). Through our mathematical modelling of an emergent, spatially spreading pathogen amongst livestock, we have demonstrated there can be a divergence in what is deemed the optimal scale for a reactive voluntary vaccination response when viewed through different social lenses (population perspective versus individual perspective). Though we recognise that the actual scenario we illustrated, of a pathogen spreading fairly rapidly between farms, is not necessarily that common in the setting of UK livestock farming, we contend it is of great importance to study such a scenario due to its potential costly negative impacts were it to arise.

Vaccination behaviour is akin to a public goods game, whereby the most effective strategy for the population is for everyone to invest in vaccination, at a personal cost. However, if a sufficient number of the population get vaccinated then 'free-riding' on others vaccination behaviour is zero-cost [32, 36, 37]. Thus, a tendency to free-ride can emerge if interventions are not decisive and/or social sanctions are imposed [38]. Therefore, minimising the cost for all the population, compared to a single individual seeking to minimise their losses, can require a broader reactive uptake of the intervention. In addition, members of the population taking an

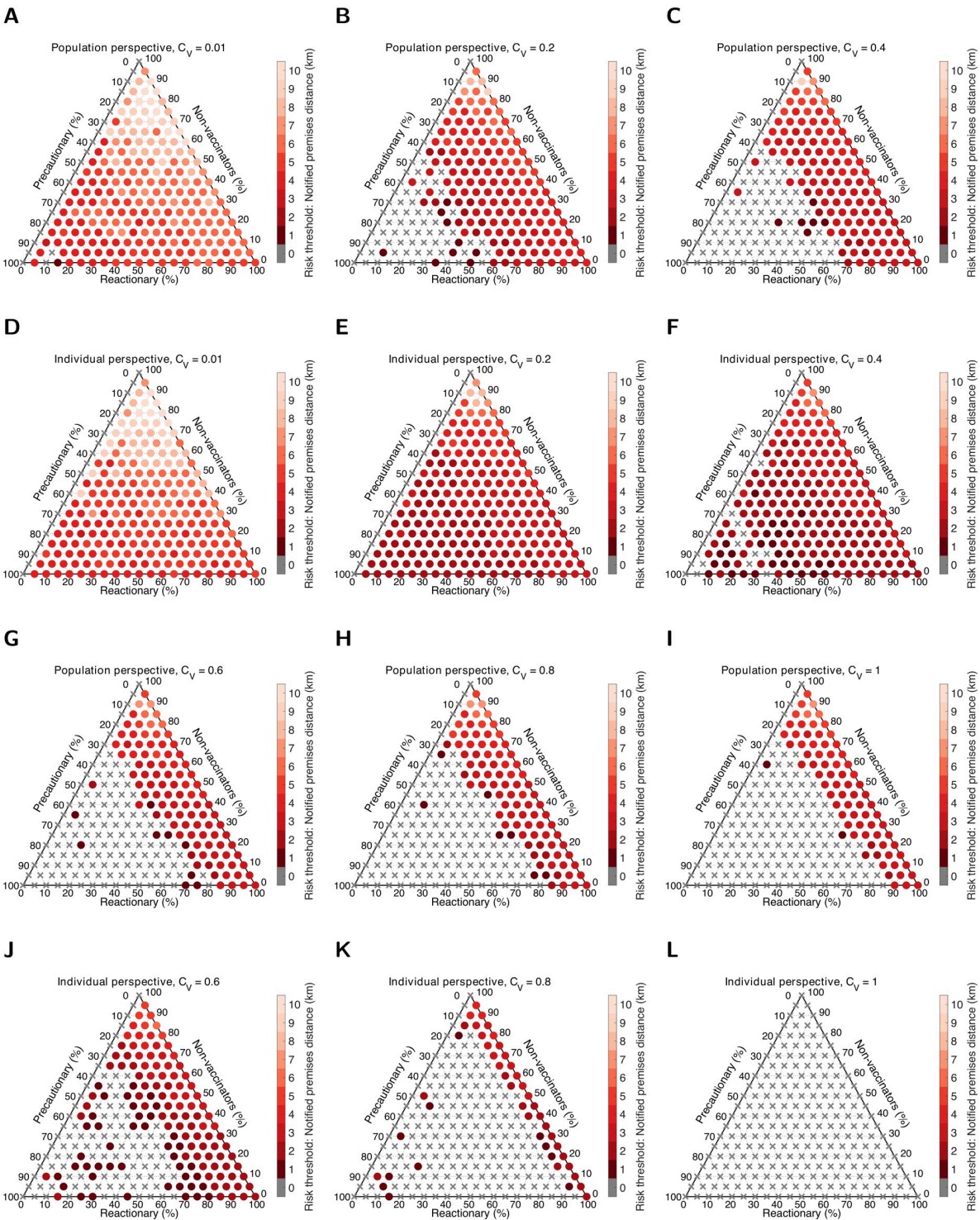

**Fig 4. Strategy that minimises overall cost in Devon, dependent on the perspective, relative cost of vaccination and vaccine group composition.** We performed 500 replicates per scenario. Shading corresponds to the notification distance risk threshold $d$ that returned the lowest median cost across the control scenarios (dark to light corresponding to increases in distance from 1km to 10km). Grey crosses correspond to no reactive vaccination occurring. The ternary plots display optimal strategies from the population and individual perspectives given a relative cost of vaccination, $C_V$, of: **A,D** 0.01; **B,E** 0.2; **C,F** 0.4; **G,J** 0.6; **H,K** 0.8; **I,L** 1. For an explanation on reading values on a ternary plot, see Fig B in the S1 Text.

individualistic approach to the control strategy adopted enhanced the prospect of larger scale disease outbreaks that were to the detriment of the population as a whole, replicating behavioural patterns observed in human vaccination behaviour [37]. In the context of public health, the COVID-19 pandemic has made clear that extinguishing infection clusters requires interventions being implemented sooner, more stringently and more broadly (spatially) than one may expect. The Scientific Advisory Group for Emergencies (SAGE) in the UK have reiterated the importance of acting early to slow a growing epidemic, with meeting minutes from the 14th October 2021 expressing "In the event of increasing case rates, earlier intervention would reduce the need for more stringent, disruptive, and longer-lasting measures" [39]. These observations raise the question within the veterinary health domain of whether farmers and national-planners should initially adopt a cautious approach in the face of an outbreak with unknown transmission potential, with a recommendation for open dialogue and clear messaging through veterinary health campaigns to culture beneficial community attitudes towards interventions. Equally, our findings indicate (for the assumed epidemiological context) where strong disagreement between the intervention stringency that is best from the perspective of a stakeholder responsible for supporting the livestock industry compared to a sole livestock owner may arise, based on the split of attitudes towards the intervention amongst the population and its relative cost compared to infection. In circumstances where the level of intervention stringency imposed on all the population would be more severe than the control response favourable to a single individual, with pushback expected from the community regarding the intervention stringency, to improve the chances of achieving a consensus across viewpoints one avenue for those managing veterinary health policy would be to lower the cost of the intervention (where possible).

In spite of these general trends, we remark that given a scenario of the majority of the farmers having vaccinated their livestock as a precautionary measure, together with the relative cost of vaccination being 0.6 or below, reactionary vaccinating farmers who wanted to minimise costs from their individual perspectives may want a broader notification zone compared to a farmer following what is deemed optimal to the overall population. A vaccine group composition predominately comprising precautionary individuals establishes a large amount of immunity across the livestock population. As a result, onward transmission from a premises is highly likely to be towards a protected premises, successfully breaking chains of transmission and meaning outbreaks are kept small. The small chance of infection entering a specific unvaccinated premises due to the herd immunity effect means, from the population perspective, vaccinating there livestock would (on average) be viewed as an unneeded cost. Indeed, people are less likely to get vaccinated if they know that sufficient others have got vaccinated [32]. On the other hand, taking an individual perspective, if there were to be a premises reporting infection nearby to an unprotected premises, the heightened risk of infection and its associated costs could make applying the intervention the cost-effective decision. These findings show that understanding individual variation in vaccination behaviour has clear implications for modelling disease outbreaks and optimising control strategies.

A pathogen seeded in different locations can exhibit dissimilar transmission dynamics. Therefore, there is room for intervention policy to be tailored at regional or local levels (if practical). Using premises spatial location and livestock demography data from Cumbria and Devon, our model simulations showed that, on average, a greater number of premises were expected to be infected in the absence of any controls in Cumbria. Thus, in our vaccination scenarios we determined Cumbria to usually require wider reactionary vaccination zone to minimise costs than Devon. That being said, there could be unintended consequences of introducing controls, with careful attention paid to the spatial expanse of their use. Tago *et al.* [40] revealed the efficacy of a movement restriction intervention policy can be substantially

reduced were it to subsequently trigger premature animal sales by farms nearby to areas under movement restrictions (thus perceiving themselves to be at high risk of becoming infected).

Swift inference of the epidemiological characteristics of a pathogen (such as its growth rate) can inform the strength of controls likely required to suppress a flare up of infection. Fundamentally, given a pathogen that has slower spreading potential (evidenced by our 'alternate' pathogen with a lowered transmissibility), the rationale would be it affords an elongated time horizon for any reactive vaccination to become effective and installing an effective ring of immunity that can act as a firebreak. Thus, the size of reactive zones implementing an intervention can be more constrained compared to pathogens with higher transmissibility. Critically, rather than improved estimates of transmission parameters, it has been found that an improved understanding of the locations of infected farms drives improved prediction of the relative performance of control interventions [41]. Accurate, reliable and robust infection surveillance systems are therefore paramount to quickly detect and eliminate new cases [42].

We have analysed a simplified model of livestock infection to illustrate basic principles related to disparities in what is deemed the optimal control policy when viewed from differing perspectives (population versus individual). Nevertheless, as with any model our use of simplifying assumptions means our results have limitations, with more work required to build our generic understanding of optimal cost-effective behaviour from the population level and the farmer perspective. First, we assumed the vaccine had 100% effectiveness (post inoculation delay). With imperfect controls the calculation of the cost from both perspectives requires additional complexity to account for the prospect of the control being ineffective. We conjecture the impact of weakened forms of intervention would hasten transmission across the premises landscape, with a need for a spatially broader reactive response as a counterbalancing measure. Second, to constrain the dimensionality of the system we streamlined the inclusion of interventions by considering a single phase vaccination control. Expansion of intervention space and the affects of other types of control (e.g. culling, increased biosecurity) were beyond the scope of this study, with further work encouraged on whether a varied control policy during an outbreak (multi-phase controls) accentuates, diminishes or leaves unaffected any discrepancies in the optimal decision between the population and individual perspectives. When using models at outbreak onset to inform policy, prior analyses have shown the importance of state-dependent interventions that adapt in response to additional information throughout an outbreak [41]. Third, we incorporated veterinary health behaviour as a single mechanism based on a perceived increasing threat of imminent infection. This mechanistic representation did not allow for individual variation in risk attitude or the influence of peers and/or influencers on a given farmer's decision to vaccinate (or not vaccinate) their livestock. Behavioural attributes are multi-faceted [43], with explicit inclusion of multiple behavioural factors in a mechanistic epidemiological-behavioural framework meriting attention, such as reinforcement (adopting use of a control as that is what your peers have done) and free-riding (not adopting a control with the belief the control having already been adopted by others will also protect you). Integrated choice and latent variable model are an option, as applied in [15] to bluetongue vaccination scheme designs, being a methodology combining structural model equations and discrete choice models that permits integration of both epidemiological and behavioural components into a single framework.

There are other open questions that warrant further study. We focused here on an epizootic situation. For management of enzootic diseases (such as bTB [44] or BVD [45]) the package of viable interventions may be bespoke and require lengthier use, which may alter the concordance in optimal strategy between the population and individual perspectives. Furthermore, with an enzootic disease there can be a regular risk of infection incursion and the potential for veterinary health behaviours to be modified after having experienced an outbreak; the model

framework would require modification to incorporate such a learning mechanism that changed a farmer's willingness to vaccinate stance. Within-premises transmission dynamics may also require consideration, an aspect not investigated here. An epidemiological-behavioural modelling framework applied by Mendes *et al.* [46], studying the control of endemic chronic diseases through privately funded actions in a low-resource setting, found the enablement of private farmers' actions to achieve a socially optimal disease control target required policy design and development that considered the heterogeneity of farmer behaviour and the predicted uptake of control measures under optimised farmer behaviour.

There are also uncertainties to explore with respect to how the optimum strategy may be affected by the difference between perceived risk (based on farms reporting cases), true risk (based on currently infected farms), and alternative perspectives that include different costs and outcomes in the analysis; parallels may be drawn from public health economics, where the point of view taken for the economic evaluation applied to a specified population can be the healthcare system (i.e. includes all costs to the health and social care system and, in terms of health outcomes, only considers health-related benefits to patients), whereas the broadest view would be a societal perspective that considers all relevant costs, whoever pays for them, including non-healthcare costs. For example, we are yet to probe how incentivisation of an intervention by a stakeholder responsible for veterinary health would alter our findings. On the basis incentivisation schemes would result in additional non-direct veterinary health related costs for a stakeholder (taking a population-level perspective on control) and a reduced intervention cost to each livestock owner, we anticipate in situations where the relative cost of the intervention is similar to the cost of infection having intervention incentives would result in a convergence in control policy irrespective of perspective taken (due to the optimal spatial range of reactive vaccination likely being: (i) reduced when evaluated from a population-level perspective; (ii) broadened when evaluated from an individual-level perspective).

Prospectively, we endeavour for strong ties between the data being gathered on farmer beliefs and the structure of epidemiological-behavioural models. It would be remiss to develop novel modelling approaches with little regard to the data available to parameterise the framework. Ensuring the data used are as reliable and accurate as possible enhances the informative capabilities and robustness of model outputs. As part of a review of livestock infectious disease models that include dynamic human behavioural change, Hidano *et al.* [13] highlighted two data challenges in veterinary epidemiology that they view as needing to be overcome to improve our understanding of human behaviour and its impact on infection in the livestock industry. First, to improve how to capture farmers' true behaviours (thus far predominantly reliant on self-reporting of practices using questionnaire surveys), they call for more rigorous qualitative methods, such as biographical narrative interviewing, and a need for quantitative studies that can capture individual, interpersonal and contextual factors (e.g. Merrill *et al.* [47] conducted an experimental simulation game to quantitatively explore the effect of interventions that would increase information sharing among stakeholders). Second, providing support for the undertaking of longitudinal studies to grow the knowledge base of how farmers' attitude, perceptions, beliefs and behaviours change over time.

In summary, we have demonstrated the capability of mathematical models that combine disease spread and socio-behavioural attributes to display anticipated differences in the livestock disease controls that minimise costs when viewed via different social perspectives. Modelling advancements that integrate epidemiological and socio-behavioural properties can help discern the scenarios, in terms of combinations of intervention receptiveness and cost, where there may be the strongest disagreement between the intervention stringency that is best for a sole individual compared to the overall population. These insights can aid our understanding of livestock owner reception to official advice during veterinary health emergencies

and indicate to bodies that devise veterinary health policy the disease management approach likely to instil unanimity with individual-level viewpoints on what is the appropriate course of disease management.

## Supporting information

**S1 Text. Includes additional figures.**
(PDF)

## Author Contributions

**Conceptualization:** Edward M. Hill, Naomi S. Prosser, Eamonn Ferguson, Jasmeet Kaler, Martin J. Green, Matt J. Keeling, Michael J. Tildesley.

**Data curation:** Edward M. Hill.

**Formal analysis:** Edward M. Hill.

**Funding acquisition:** Eamonn Ferguson, Jasmeet Kaler, Martin J. Green, Matt J. Keeling, Michael J. Tildesley.

**Methodology:** Edward M. Hill, Matt J. Keeling, Michael J. Tildesley.

**Software:** Edward M. Hill.

**Supervision:** Eamonn Ferguson, Jasmeet Kaler, Martin J. Green, Matt J. Keeling, Michael J. Tildesley.

**Validation:** Edward M. Hill.

**Visualization:** Edward M. Hill, Matt J. Keeling.

**Writing – original draft:** Edward M. Hill.

**Writing – review & editing:** Edward M. Hill, Naomi S. Prosser, Eamonn Ferguson, Jasmeet Kaler, Martin J. Green, Matt J. Keeling, Michael J. Tildesley.

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
