## [Decision Letter · Decision Letter 0]

12 Apr 2022

Dear Dr Hill,

Thank you very much for submitting your manuscript "Modelling livestock infectious disease control policy under differing social perspectives on vaccination behaviour" for consideration at PLOS Computational Biology. As with all papers reviewed by the journal, your manuscript was reviewed by members of the editorial board and by several independent reviewers. The reviewers appreciated the attention to an important topic. Based on the reviews, we are likely to accept this manuscript for publication, providing that you modify the manuscript according to the review recommendations.

In particular, I ask that you:

 - consider Reviewer #1's comment on the choice of kernel. At a minimum, the point identified by the reviewer should be acknowledged and discussed in the text.

 - consider improvements to the abstract and main text conclusions to address Reviewer #2's comment on implications and meaning that can be drawn from this work.

Sincerely,

James M McCaw, PhD

Associate Editor

PLOS Computational Biology

Virginia Pitzer

Deputy Editor-in-Chief

PLOS Computational Biology

[LINK]

Reviewer's Responses to Questions

**Comments to the Authors:**

Reviewer #1: The authors tried to develop a parsimonious modelling framework to combine decision dynamics with epidemiological dynamics. Classically these dynamics are often separately assessed and feed-back ignored. The authors base their efforts on foot-and-mouth outbreaks in two English counties specifying three types of farmers: precautionary, reactionary and non-vaccinators.

The conclusions are substantiated by the results and extensive sensitivity analysis of their model. This of course then relies on the suitability of the model to study the given dynamics.

Intentionally the model is kept fairly simple allowing for robust and tractable results. This illustrates under the model assumptions the potential disagreement between population level and individual level perspective on the most optimal strategy.

The optimal decision to vaccinate for reactionary vaccinators is solely based on the distance to infected farms and costs for vaccination and infection. This requires a full knowledge of the epidemiological state of other farms as well as knowledge on the transmission kernel. Both are often not available during (or directly after) an outbreak. It also does not allow for individual variation in risk attitude.

Another draw-back of this approach is that the model does not take into account the influence of peers or influential people on the decision to vaccinate. Non-vaccinators remain non-vaccinators in the model even when all their peers do vaccinate and might potentially pressure non-vaccinators to vaccinate. Also reactionary vaccinators might be persuaded to act differently with many non-vaccinators in their peer-network.

For an epizootic infections such as FMD in Europe, learning might not play a role, but the framework does not allow for changes in willingness to vaccinate after having experienced outbreaks. The authors acknowledge this limitation in the discussion.

This modelling framework is, altogether, an important addition in the toolbox of modelling for evaluation of intervention strategies.

Some minor remarks remain:

- Why choose for a parameterization of the kernel based on US outbreak? The transmission kernel represents both the infectivity of a particular strain as the contact structure within a country, which is likely to be very different in UK and US.

Reviewer #2: The abstract needs strengthening (beyond ‘deepening our understanding’) in terms of providing recommendations based on the authors findings:

By identifying instances of strong disagreement between the intervention stringency that is best from the perspective of a sole individual compared to the overall population, we can deepen our understanding of how stakeholders may react to veterinary health interventions.

• This is a timely paper, as noted. There has been a call by the infectious disease modelling community, in the human public health field, for greater incorporation of behavioural changes among infectious disease dynamic models.

• There is no mention of the role of incentives given the authors findings. Incentives are likely to be an important discussion point. This should be built into the paper

• From an economic point of view the significance of the differing perspectives (esp the population) in this context needs to be developed (e.g. In other areas of economics, such as health economics, perspective such as the NHS and Societal are relevant.)

The authors note: Population perspective The population perspective took the viewpoint of an entity who wanted to minimise costs for the overall population. Can this be expanded? For example what is the ‘entity’ and what are the relevant incentives for such an entity?

• Conclusions and recommendations need much more development based on the findings.

**Have the authors made all data and (if applicable) computational code underlying the findings in their manuscript fully available?**

Reviewer #1: Yes

Reviewer #2: Yes

PLOS authors have the option to publish the peer review history of their article (what does this mean?). If published, this will include your full peer review and any attached files.

Reviewer #1: **Yes: **Egil Fischer

Reviewer #2: No

Figure Files:

Data Requirements:

Reproducibility:

References:

---

## [Editor Report · Decision Letter 1]

20 May 2022

Dear Dr Hill,

We are pleased to inform you that your manuscript 'Modelling livestock infectious disease control policy under differing social perspectives on vaccination behaviour' has been provisionally accepted for publication in PLOS Computational Biology.

Best regards,

James M McCaw, PhD

Associate Editor

PLOS Computational Biology

Virginia Pitzer

Deputy Editor-in-Chief

PLOS Computational Biology

---

## [Editor Report · Acceptance letter]

23 Jun 2022

PCOMPBIOL-D-22-00223R1 

Modelling livestock infectious disease control policy under differing social perspectives on vaccination behaviour

Dear Dr Hill,

I am pleased to inform you that your manuscript has been formally accepted for publication in PLOS Computational Biology. Your manuscript is now with our production department and you will be notified of the publication date in due course.

With kind regards,

Zita Barta
